# Semi-Markov Model of the System of Repairs and Preventive Replacements by Age of City Buses

**Klaudiusz Migawa** 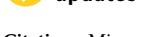, **Sylwester Borowski \*, Andrzej Neubauer and Agnieszka Sołtysiak**

Faculty of Mechanical Engineering, UTP University of Science and Technology, Al. Prof. S. Kaliskiego 7, 85-796 Bydgoszcz, Poland; klaudiusz.migawa@pbs.edu.pl (K.M.); aneub@umk.pl (A.N.); agnieszka.soltysiak@pbs.edu.pl (A.S.)
* Correspondence: sylwester.borowski@pbs.edu.pl

**Abstract:** The paper presents a mathematical model of the system of repairs and preventive replacements by age of city buses. The mathematical model was developed using the theory of semi-Markov processes. In the model developed, four types of city bus renewal processes are considered and three types of corrective repairs and preventive replacement. Corrective repairs are considered in two types: minimal repairs (repairs carried out by the Technical Service units) and perfect repairs (repairs carried out at the stations of the Service Station). The models of restoration systems that use semi-Markov processes in which minimal repairs, perfect repairs, and preventive replacements by age, have been examined in the literature to a limited extent. The system under consideration is analysed from the point of view of two criteria: profit per time unit and availability of city buses to carry out the assigned transport tasks. Conditions of criterion functions' extremum (maximum) existence were formulated for the adopted assumptions. The considerations presented in the paper are illustrated by exemplary results of calculations.

**Keywords:** city buses; semi-Markov processes; preventive maintenance; corrective maintenance; age-replacement; minimal repair; perfect repair; profit per time unit; availability

## 1. Introduction

The basic task of transport systems is to transport people, animals and goods. The transport tasks, due to their particular specification, are carried out by different types of transport means. A very important branch of the transport system is the road passenger transport, which can generally be divided into international transport, interurban transport and urban transport. Urban transport systems usually operate in medium and large cities, in suburban areas and in industrialised areas. One of the important types of urban transport is the urban bus transport. The task of this type of transport system is to reliably and punctually carry out transport tasks along defined routes in accordance with an accepted timetable of courses [1] The basic characteristics for evaluating the functioning of this type of transport system are economic efficiency characteristics (e.g., profit per unit time) and operational-technical efficiency characteristics (e.g., readiness of the city buses to carry out the assigned transport tasks) [2,3]. Any kind of disruption in the implementation of the assigned transport tasks, including downtime caused by damages to the means of transport (city buses) causes a decrease in the reliability and readiness of technical facilities and generates additional costs (losses). These losses arise as a result of corrective maintenance (CM) conducted after damage, losses caused by fines for non-performance of transport services, and costs related to the maintenance of reserve buses whose task is to replace damaged buses. One of the ways of ensuring the correct and efficient fulfilment of transport tasks in urban bus operation systems is the implementation of preventive maintenance (PM). The implementation of these activities consists of planning the timing and scope of preventive maintenance in such a way as to keep their costs lower than the costs of repairs after the damage. For this reason, the determination of optimal times for

preventive maintenance of technical objects is an important problem of planning strategies in the systems of exploitation of the means of transport [4].

During the operation of technical objects, their elements are subject to wear and tear processes and the impact of external factors, which causes damage to these objects. The resulting damages are the cause of lowering the effectiveness of functioning of the analysed systems. In order to ensure an appropriate level of reliability of technical facilities, different types of strategies are applied in the subsystems for ensuring serviceability. These activities are divided into two types: preventive maintenance and corrective maintenance. In practice, corrective maintenance is carried out in two variants. Firstly, the perfect repair (PR), which makes the system "As-Good-As-New" (AGAN), and secondly the minimal repair (MR), which makes the system "As-Bad-As-Old" (ABAO) are conducted. Normally, CM corrective repair costs and times are higher than PM preventive repair costs. This is due to the fact that, in general, CM activities require prior identification of the damage and high skills of personnel (diagnosticians and mechanics). In addition, there are the costs associated with the unplanned downtime of technical facilities caused by the damage to them. From this, it follows that it is possible to plan preventive actions (the scope and frequency of preventive repairs and replacements) in such a way as to ensure the required level of readiness of technical facilities and to reduce system maintenance costs. This requires the development and application of rational preventive repair and replacement strategies. For these reasons, the development of various preventive action strategies with the application of optimal decision models to reduce the system maintenance costs and the risk of adverse events is an important research topic in reliability engineering.

For the first time, the concept of minimal repair can be found in Morse's study [5]. In this study, a repair model is considered in which the criterion function is the monthly revenue generated for the technical facility under consideration. However, this model was developed on the basis of the queueing theory and not the reliability theory. However, the concept of minimum repairs in relation to the reliability theory was introduced by Barlow and Hunter in their paper [6]. In this paper, the model of periodic replacements and minimum repairs is considered, in which it is assumed that after each minimum repair the damaged technical system is restored only to the same failure condition as before the damage. In a formal way, the concept of minimum repairs was defined by Nakagawa and Kowada in their paper [7]. In the paper [8] Brown and Proschan also consider the issue of minimal repair. In this paper, the authors assume that when a technical object is damaged, a perfect repair is performed with probability p, while a minimal repair is performed with probability q = 1 − p. A modified version of such a model was proposed by Fontenot and Proschan [9]. In the model developed, the object is replaced with a new one after time T and either a perfect repair or a minimal repair is performed with probabilities p and q, respectively, for intermittent failures.

In the literature one can find descriptions of models of systems with minimum repairs, which have been developed with the use of various methods and mathematical tools. An overview of the used modelling methods and the construction of criterion functions in models of minimum repairs with preventive maintenance can be found, for example, in the papers [10,11]. The papers classify and discuss models of maintenance strategies for technical objects developed for both finite and infinite time horizons, in which the criterion functions are total costs, unit costs, reliability and readiness. Most of the models presented in the literature have been developed on the basis of renewal theory, while less frequently with the use of stochastic processes, including Markov and semi-Markov process models. For example, in the paper [12] the criterion functions cost per unit time and system availability were determined on the basis of a semi-Markov model in an infinite time horizon, and in the paper [13] the model of the imperfect maintenance system was developed using the theory of Markov processes, and the readiness function is a criterion for optimisation.

In practice, the effectiveness of the realised repair is between AGAN and ABAO repair and it concerns the so-called imperfect maintenance/repairs. The methods concerning

preventive repairs and replacements using the repair mechanism with an imperfect mainte-nance model with the (p, q) rule are extensively discussed in the paper [10]. The paper [14] presents the problem of imperfect repair with periodic preventive replacement. Models of preventive replacements by age are presented in papers [15,16]. In this type of model, it was assumed that the probabilities p and q depend on the age of the technical object at the time of failure, and that a thorough repair restores the technical object to the reliability state as for a new object, while a minimal repair restores the technical object to the reliability state just before failure. In the paper [17] it was shown that the PM policy limiting the possibility of failure can be more cost-effective than the PM policy implemented according to age, while the authors of the papers [18,19] analysed the imperfect repair system model with a delayed time concept.

Models of imperfect maintenance systems, using different age replacement policies that take into account different types of repairs after failure and their cost structures, have been presented in a number of papers. In [20], the authors consider replacement policies depending on the age of the system and the minimisation of repair costs. The authors of the papers [21,22] consider the age replacement policy of system subject to shocks in their models. Other papers consider policies that assume randomness of model parameters, e.g., papers [23,24] assume random repair costs. On the other hand, the paper [25] describes an age replacement policy with Bayesian imperfect repair model, in which the probability of an exact repair is a random variable with a specified distribution. A similar approach is adopted in the paper [26], where the optimal age replacement policy is determined in the case of minimising the cost per unit time. The results were obtained both for an infinite time horizon and for a single replacement cycle. The sequential imperfect preventive maintenance model for city buses is presented in the paper [27]. In this model, the optimal decision-making concerning the efficiency of maintenance of city buses is realized on the basis of the evaluation of the difference between the actual and expected increments of the intensity of damage.

The results presented in this paper are a continuation of the considerations presented in papers [4,28,29]. Similarly, to this paper, the results presented in these papers were obtained based on the study of the semi-Markov models. In the paper [4] a 4-state model of replacements according to the age of technical objects with a guarantee was analysed, in which the criterion function is the cost of preventive replacement determined per unit of time, while in the paper [28] a multi-state model of exploitation decisions was developed, in which corrective repairs (after a damage) and preventive replacements are carried out, and the profit per unit of time was used as the criterion function. A direct continuation of the conducted research are the results presented in the paper [29], in which a 4-state model of a service system with minimum repair was considered. The model was developed with the application of the theory of semi-Markov processes, and the theoretical considerations were illustrated with numerical examples on the basis of the assumed sample data. In this paper, on the other hand, the 5-state semi-Markov model of the system of preventive repairs and replacements according to the age of city buses is considered. The aim of the paper is to develop and study a semi-Markov model of preventive repairs and replacements according to the age of city buses using two criterion functions: profit per unit time and the coefficient of readiness of city buses to carry out the assigned transport tasks, and also to formulate sufficient conditions for the existence of the maximum of these functions. The results of studies of the model, developed on the basis of real data, can be the basis for decision-making in the analysed system of exploitation of urban transport means. In the developed mathematical model, the basis for the construction of the criterion function is the limit theorem for semi-Markov processes [30,31]. In the developed model, four types of implemented urban bus renewal processes are considered. The conditions for the existence of the extremum (maximum) of the criterion functions were formulated for the assumed assumptions. The theoretical considerations presented in the paper are illustrated by the results of calculations. The calculation examples have been developed on the basis of operational data obtained from a real urban bus operation system. In the first example, for

the estimated input data, the profit per unit time and the value of the readiness factor are maximised. In the second example, the analysed criterion functions are investigated, in case when the number (frequency) of preventive replacements will be higher than in the first example. In both the calculation examples, it is assumed that the time to failure of the technical object (city bus) has a Weibull distribution.

## 2. Description of the States of the Model of the Repair and Preventive Replacement System

In the study the object of research is the bus exploitation system of public transport, in which technical objects (city buses) can stay in one of the five states of the considered model of the renewal system (repairs and preventive replacements):

State 1—the state of operational availability of the technical object—it is the state, when the technical object (a city bus) is fully fit and supplied and can perform the assigned transport task in accordance with the accepted assumptions, i.e., in accordance with the accepted plan and schedule of realization of the courses (in the paper this state is considered as the state of failure-free work of the technical object);

State 2—the state of repair by the Technical Rescue Service without the loss of the course—this is the state, when a damaged technical object (a city bus) is repaired during the realization of the transport task (on the route), which is carried out by the Technical Rescue Service; this type of repair is carried out in a "short" time interval or in the gaps between the successive realizations of the transport task (between the next courses), i.e., it does not cause loss of the course in accordance with the accepted plan and schedule of transport realization—it is assumed that this condition does not cause any disturbance (break) in realization of assigned transport tasks (in the paper this condition is considered as a condition of minimal repair of a technical object);

State 3—the state of the repair by the Technical Rescue Unit with the loss of the course—it is the state, when a damaged technical object (a city bus) is repaired during the realization of the transport task (on the route), which is realized by the Technical Rescue Unit; this type of repair is realized in a "longer" time interval than the repair realized in the state 2 of the model, thus the repair causes the loss of the course in accordance with the plan and schedule of transportation realization—it is assumed that this state causes a disruption (break) in the realization of the assigned transportation tasks (in the paper this state is considered as the state of the minimal repair of the technical object);

State 4—state of repair at the Service Station—it is the state, when a damaged technical object (a city bus) is subject to repair at the specialized service and repair stands of the Service Station assigned for this purpose—it is assumed that this state causes a disturbance (break) in the realization of the assigned transport tasks (in the paper, this state is considered as the state of perfect repair of a technical object);

State 5—the state of preventive replacement—it is the state, when the technical object (a city bus) is subject to preventive maintenance after a specific hourly mileage and in accordance with the adopted operational strategy (according to the resour) in the technical object the elements and sub-assemblies are replaced—it is assumed, that this state does not cause disruption (break) in the realization of the assigned transport tasks (in the work this state is considered as the state of preventive age-replacement.).

Figure 1 shows a directed graph of the mapping of the state changes of the renewal system model (preventive repairs and replacements) of the considered technical objects (city buses).

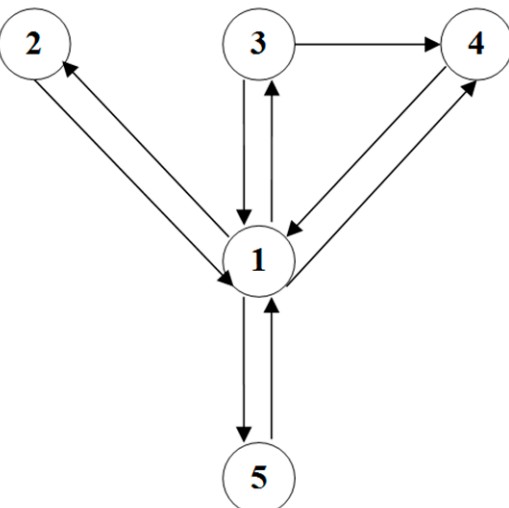

**Figure 1.** Directed graph representation of the state changes of a city bus renewal system model with state spaces S = {1, 2, 3, 4, 5}.

### 3. Determination of the Criterion Function

For the directed graph shown in Figure 1, a mathematical model was built assuming that it is a stochastic process X(t). The mathematical model was developed using the theory of semi-Markov processes [30,31]. The paper considers a 5-state semi-Markov model of renewals (preventive repairs and replacements) with a state space S = {1, 2, 3, 4, 5}. If X(t) = i, then the technical object under consideration at time t is in state i.

In the case when the transition probabilities between the states of the modelled process are known, it is possible to determine the Markov chain inserted in the semi-Markov process. The transition matrix of the Markov chain for the model under consideration has the form

$$
P = \begin{bmatrix}
0 & P_{12} & P_{13} & P_{14} & P_{15} \\
P_{21} & 0 & 0 & 0 & 0 \\
P_{31} & 0 & 0 & P_{34} & 0 \\
P_{41} & 0 & 0 & 0 & 0 \\
P_{51} & 0 & 0 & 0 & 0
\end{bmatrix}
\tag{1}
$$

where

$p_{ij}$, i, j = 1, 2, 3, 4, 5—probability of transition from state i to state j.

To determine the limiting probabilities for a Markov chain, the following matrix system must be solved:

$$
P^T \cdot \Pi = \Pi \begin{bmatrix}
0 & P_{21} & P_{31} & P_{41} & P_{51} \\
P_{12} & 0 & 0 & 0 & 0 \\
P_{13} & 0 & 0 & 0 & 0 \\
P_{14} & 0 & P_{34} & 0 & 0 \\
P_{15} & 0 & 0 & 0 & 0
\end{bmatrix} \cdot \begin{bmatrix}
\pi_1 \\ \pi_2 \\ \pi_3 \\ \pi_4 \\ \pi_5
\end{bmatrix} = \begin{bmatrix}
\pi_1 \\ \pi_2 \\ \pi_3 \\ \pi_4 \\ \pi_5
\end{bmatrix}
\tag{2}
$$

where

$\pi_i$, i = 1, 2, 3, 4, 5—the limiting probability of a Markov chain inserted in a semi-markov process.

The Matrix System (2) can be replaced by a system of linear Equation (4) in which, in order to obtain an unambiguous solution, a normalization condition is introduced (3)

$$
\sum_i \pi_i = 1
\tag{3}
$$

then the system of linear Equation (4) takes the form

$$
\begin{cases}
\pi_1 + \pi_2 + \pi_3 + \pi_4 + \pi_5 = 1 \\
p_{12} \cdot \pi_1 = \pi_2 \\
p_{13} \cdot \pi_1 = \pi_3 \\
p_{14} \cdot \pi_1 + p_{34} \cdot \pi_3 = \pi_4 \\
p_{15} \cdot \pi_1 = \pi_5
\end{cases}
\tag{4}
$$

As a result of solving the system of linear Equation (4), formulas representing the limiting probabilities for the analysed Markov chain were obtained:

$$
\pi_1 = \frac{1}{m} \, \pi_2 = \frac{p_{12}}{m} \, \pi_3 = \frac{p_{13}}{m} \, \pi_4 = \frac{p_{13} \cdot p_{34} + p_{14}}{m} \, \pi_5 = \frac{p_{15}}{m}
\tag{5}
$$

where

$$
m = 1 + p_{12} \cdot p_{13} \cdot (1 + p_{34}) + p_{14} + p_{15}
$$

In this paper, a semi-Markov model of renewals (preventive corrective maintenance and age-replacement) is analysed. A 5-state semi-Markov process $X(t)$ with state space $S = \{1, 2, 3, 4, 5\}$ is considered. By $z_i$, $i = 1, 2, 3, 4, 5$ means the profit (cost) per unit (per time unit) for the state i. It is assumed in the paper, that $z_1 > 0$, $z_i < 0$ for $i = 2, 3, 4, 5$. This means that if the technical object is in state 1, then a profit is generated, while if the technical object is in state $i = 2, 3, 4, 5$, then a cost (loss) is generated. In the paper [28] it was proved, that the summary profit (loss) per unit of time generated in the system is expressed by the formula

$$
Z = \frac{\sum_i \pi_i \cdot ET_i \cdot z_i}{\sum_i \pi_i \cdot ET_i}
\tag{6}
$$

where

$ET_i$, $i = 1, 2, 3, 4, 5$–average time spent in state i.

A technical object is subject to renewal at age T or when it is damaged, whatever comes first. By $T_1(x)$ we define the time to replace or damage (repair) a technical object. The variable $T_1(x)$ is defined as follows

$$
T_1(x) = \begin{cases}
T_1, & \text{gdy} \quad T_1 < x \\
x, & \text{gdy} \quad T_1 \geq x
\end{cases}
\tag{7}
$$

It is assumed that after time x, if the technical object has not failed, it transitions to the preventive replacement state. The process of changing states $i = 1, 2, 3, 4, 5$, given preventive replacement after time x is a new semi-Markov process with a matrix $P(x)$ of transition probabilities of the Markov chain inserted into the semi-Markov process. With respect to the matrix $P(1)$ shown above, only the first row of the matrix P changes, then the matrix $P(x)$ takes the form

$$
P(x) = \begin{bmatrix}
0 & p_{12}(x) & p_{13}(x) & p_{14}(x) & p_{15}(x) \\
p_{21} & 0 & 0 & 0 & 0 \\
p_{31} & 0 & 0 & p_{34} & 0 \\
p_{41} & 0 & 0 & 0 & 0 \\
p_{51} & 0 & 0 & 0 & 0
\end{bmatrix}
\tag{8}
$$

while the limiting probabilities determined for a Markov chain (determined analogously to Formulae (5) can be presented as:

$$\pi_1(x) = \tfrac{1}{n}$$
$$\pi_2(x) = \tfrac{p_{12}(x)}{n}$$
$$\pi_3(x) = \tfrac{p_{13}(x)}{n} \tag{9}$$
$$\pi_4(x) = \tfrac{p_{13}(x) \cdot p_{34} + p_{14}(x)}{n}$$
$$\pi_5(x) = \tfrac{p_{15}(x)}{n}$$

where
$$n = 1 + p_{12}(x) \cdot p_{13}(x) \cdot (1 + p_{34}) + p_{14}(x) + p_{15}(x)$$

Based on the paper [28], the Criterion Function (6) is of the form:

$$Z = g(x) = \frac{\pi_1(x) \cdot ET_1(x) \cdot z_1 + \pi_2(x) \cdot ET_2 \cdot z_2 + \pi_3(x) \cdot ET_3 \cdot z_3 + \pi_4(x) \cdot ET_4 \cdot z_4 + \pi_5(x) \cdot ET_5 \cdot z_5}{\pi_1(x) \cdot ET_1(x) + \pi_2(x) \cdot ET_2 + \pi_3(x) \cdot ET_3 + \pi_4(x) \cdot ET_4 + \pi_5(x) \cdot ET_5} \tag{10}$$

where
$ET_1(x)$–average value of time spent in state 1, calculated from the formula [30,31]

$$ET_1(x) = \int_0^x dF_1(T) + xP\{T_1 \geq x\}$$
$$ET_1(x) = \int_0^x R_1(T)dT \tag{11}$$

$ET_2$, $ET_3$, $ET_4$ and $ET_5$,—average values of times spent in states 2, 3, 4 and 5, respectively. In particular, based on the paper [28], it can be written:

$$p_{12}(x) = p_{12} \cdot F_{12}(x)$$
$$p_{13}(x) = p_{13} \cdot F_{13}(x)$$
$$p_{14}(x) = p_{14} \cdot F_{14}(x) \tag{12}$$
$$p_{15}(x) = p_{15} \cdot F_{12}(x) + R_1(x)$$

where:
$F_{1j}(x)$, j = 2, 3, 4, 5—conditional distributions of the time spent in state 1, provided that the next state is state j, defined as follows [30,31]

$$F_{ij}(t) = P\{\tau_{k+1} - \tau_k < t | X(\tau_{k+1}) = j, X(\tau_k) = i\}, \text{ dla } i, j = 1, 2, 3, 4, 5 \tag{13}$$

$R_1(x) = 1 - F_1(x)$—random variable reliability function $T_1$.

Additionally, in order to simplify further considerations, it has been assumed that the following equations are true

$$F_{12}(x) = F_{13}(x) = F_{14}(x) = F_{15}(x) = F_1(x) \tag{14}$$

Considering the above, the Criterion Function (10) is expressed by the formula

$$g(x) = \frac{\begin{aligned}&ET_1(x) \cdot z_1 + p_{12} \cdot F_1(x) \cdot ET_2 \cdot z_2 + p_{13} \cdot F_1(x) \cdot ET_3 \cdot z_3 + \left[(p_{13} \cdot p_{34} + p_{14}) \cdot F_1(x)\right] \cdot ET_4 \cdot z_4 + \\ &+ \left[1 - (p_{12} + p_{13} + p_{14}) \cdot F_1(x)\right] \cdot ET_5 \cdot z_5\end{aligned}}{\begin{aligned}&ET_1(x) + p_{12} \cdot F_1(x) \cdot ET_2 + p_{13} \cdot F_1(x) \cdot ET_3 + \left[(p_{13} \cdot p_{34} + p_{14}) \cdot F_1(x)\right] \cdot ET_4 + \\ &+ \left[1 - (p_{12} + p_{13} + p_{14}) \cdot F_1(x)\right] \cdot ET_5\end{aligned}}$$

or after regrouping, it can be represented as

$$g(x) = \frac{z_1 \cdot ET_1(x) + \left[p_{12} \cdot ET_2 \cdot z_2 + p_{13} \cdot ET_3 \cdot z_3 + \left(p_{13} \cdot p_{34} + p_{14}\right) \cdot ET_4 \cdot z_4 +}{ET_1(x) + \left[p_{12} \cdot ET_2 + p_{13} \cdot ET_3 + \left(p_{13} \cdot p_{34} + p_{14}\right) \cdot ET_4 +}$$

$$\frac{- \left(p_{12} + p_{13} + p_{14}\right) \cdot ET_5 \cdot z_5\right] \cdot F_1(x) + ET_5 \cdot z_5}{- \left(p_{12} + p_{13} + p_{14}\right) \cdot ET_5\right] \cdot F_1(x) + ET_5}$$

Representing the numerator and the denominator of the criterion function as follows:

$$L(x) = A_1 \cdot ET_1(x) + B_1 \cdot F_1(x) + C_1$$

$$M(x) = A \cdot ET_1(x) + B \cdot F_1(x) + C$$

the Criterion Function (10) can be represented analogously as

$$g(x) = \frac{A_1 \cdot ET_1(x) + B_1 \cdot F_1(x) + C_1}{A \cdot ET_1(x) + B \cdot F_1(x) + C}$$

where:

$$A_1 = z_1$$

$$B_1 = p_{12} \cdot ET_2 \cdot z_2 + p_{13} \cdot ET_3 \cdot z_3 + \left(p_{13} \cdot p_{34} + p_{14}\right) \cdot ET_4 \cdot z_4 - \left(p_{12} + p_{13} + p_{14}\right) \cdot ET_5 \cdot z_5$$

$$C_1 = ET_5 \cdot z_5$$

$$A = 1$$

$$B = p_{12} \cdot ET_2 + p_{13} \cdot ET_3 + \left(p_{13} \cdot p_{34} + p_{14}\right) \cdot ET_4 - \left(p_{12} + p_{13} + p_{14}\right) \cdot ET_5$$

$$C = ET_5$$

## 4. Conditions for the Existence of a Maximum of the Criterion Function

### 4.1. Maximum of the Criterion Function—General Analysis

The conditions for the existence of an extremum (maximum) of the Criterion Function (10) will be formulated depending on the parameters of the developed semi-Markov model of the renewal system (repairs and preventive replacements), i.e., the elements of the matrix of probabilities of changes in the states of the model $P = [p_{ij}]$, $i, j = 1, 2, 3, 4, 5$, the average staying times in the states of the model $ET_i$, $i = 1, 2, 3, 4, 5$, and the unit profits (costs) generated in the states of the model $z_i$, $i = 1, 2, 3, 4, 5$. The considered parameters are the input data of the model, and their values depend on the category and type of the analysed technical objects, the adopted operation strategy, and specific operating conditions in which the repair and preventive replacement processes are carried out.

The assumptions regarding the values of the parameters of the examined system are defined below. The adopted assumptions must take into account the real relations occurring between the parameters characterizing the implemented processes of repair of the damaged technical objects and the preventive replacement:

- Z1: $z_1 > 0$, $z_2 < 0$, $z_3 < 0$, $z_4 < 0$, $z_5 < 0$; means that a technical object staying in state 1 generates profit (+), whereas staying in states 2, 3, 4 and 5 generates costs (–);
- Z2: $ET_2 < ET_3$; means, that the value of the average time of repair carried out by the Technical Service units without the loss of the course is lower than the value of the average time of repair carried out by the Technical Service units with the loss of the course;
- Z3: $ET_2 < ET_4$; means that the value of the average repair time performed by the Technical Service units without the loss of the course (minimal repair) is lower than the value of the average repair time performed at the Service Stations (perfect repair));
- Z4: $ET_3 < ET_4$; means that the value of the average repair time realized by the Technical Service units with the loss of the course (minimal repair) is lower than the value of the average repair time realized at the Service Stations (perfect repair);

- Z5: $ET_4 > ET_5$; means that the value of the average time of repair performed at Service Stations (precise repair) is higher than the value of the average time of preventive replacement;
- Z6: $ET_2 < ET_5$; means, that the value of the average time of repair carried out by the Technical Service units without the loss of the course (minimal repair) is lower than the value of the average time of preventive replacement;
- Z7: $z_2 < z_3$; means, that the unit cost generated in the state 2 (the state of repair realized by the Technical Service units without the loss of the course) is lower than the unit cost generated in the state 3 (the state of repair realized by the Technical Service unit with the loss of the course);
- Z8: $z_2 < z_4$; means, that the unit cost generated in the state 2 (the state of the repair carried out by the Technical Service units without the loss of the course) is lower than the unit cost generated in the state 4 (the state of the repair carried out at the Service Stations);
- Z9: $z_3 < z_4$; means, that the unit cost generated in the state 3 (the state of the repair carried out by the Technical Service units with the loss of the course) is lower than the unit cost generated in the state 4 (the state of the repair carried out at the Service Stations);
- Z10: $z_4 > z_5$; means, that the unit cost generated in the state 4 (the state of repair performed at the Service Stations) is higher than the unit cost generated in the state 5 (the state of preventive replacement);
- Z11. $z_2 < z_5$; means, that the unit cost generated in the state 2 (the state of repair carried out by the Technical Service units without the loss of the course) is lower than the unit cost generated in the state 5 (the state of preventive replacement).

The above assumptions do not define the relationship between the state of repair performed by the Technical Service unit with loss of the course (state 3) and the state of preventive replacement (state 5). In the analysed system it is very difficult to unambiguously define the relation between the average values of ET3 and ET5 times and unit costs z3 and z5. However, based on the results of research on other systems of exploitation of this class of technical objects (means of transport), an additional assumption can be made regarding the unit costs generated in states 3 and 5:

- Z12: $z_3 < z_5$; means that the unit cost generated in the state 3 (the state of repair performed by the Technical Service unit with the loss of the course) is lower than the unit cost generated in the state 5 (the state of preventive replacement).

In the subsequent part of the paper, the following coefficients have been introduced to formulate the conditions for the existence of the extremum (maximum) of the criterion function (10):

$$\alpha = AB_1 - A_1B = B_1 - z_1B$$

$$\beta = A_1C - AC_1 = z_1C - C_1$$

$$\gamma = B_1C - BC_1$$

where:

$$A_1 = z_1$$

$$B_1 = p_{12}ET_2z_2 + p_{13}ET_3z_3 + (p_{13}p_{34} + p_{14})ET_4z_4 - (p_{12} + p_{13} + p_{14})ET_5z_5$$

$$C_1 = ET_5z_5$$

$$A = 1$$

$$B = p_{12}ET_2 + p_{13}ET_3 + (p_{13}p_{34} + p_{14})ET_4 - (p_{12} + p_{13} + p_{14})ET_5$$

$$B = p_{12}ET_2 + p_{12}p_{23}ET_3 - p_{12}ET_4$$

$$C = ET_5$$

The coefficients $\alpha$, $\beta$, and $\gamma$ play an important role in formulating sufficient conditions for the existence of the extremes of a criterion function. For this purpose, the following formulates the sufficient conditions for the inequalities to be true $\alpha < 0$, $\beta > 0$, $\gamma < 0$.

In regard to the above:

- the coefficient $\alpha$ is determined by the formula

$$\alpha = p_{12}ET_2(z_2 - z_1) + p_{13}ET_3(z_3 - z_1) + (p_{13}p_{34} + p_{14})ET_4(z_4 - z_1) + (p_{12} + p_{13} + p_{14})ET_5(z_1 - z_5) \tag{15}$$

The inequality $\alpha < 0$ is equivalent to the inequality

$$(p_{13} + p_{34} + p_{14}) \quad ET_4 > p_{12}ET_2(z_2 - z_1)/(z_1 - z_4) + p_{13}ET_3(z_3 - z_1)/(z_1 - z_4) + (p_{12} + p_{13} + p_{14})ET_5(z_1 - z_5)/(z_1 - z_4) \tag{16}$$

- the coefficient $\beta$ is determined by the formula

$$\beta = ET_5(z_1 - z_5) \tag{17}$$

Based on the assumption Z1 made, it follows that $\beta > 0$.

- the coefficient $\gamma$ is determined by the formula

$$\gamma = \left[ p_{12}ET_2(z_2 - z_5) + p_{13}ET_3(z_3 - z_5) + (p_{13}p_{34} + p_{14})ET_4(z_4 - z_5) \right]ET_5 \tag{18}$$

The inequality $\gamma < 0$ is equivalent to the inequality

$$(p_{13} + p_{34} + p_{14})ET_4 > p_{12}ET_2(z_2 - z_5)/(z_5 - z_4) + p_{13}ET_3(z_3 - z_5)/(z_5 - z_4) \tag{19}$$

In practice, it is difficult to unambiguously determine what is the relation between the state of repair by the Technical Service unit with course loss (state 3) and the state of preventive replacement (state 5), i.e., the relation between the average values of the times $ET_3$ and $ET_5$ and between the unit costs $z_3$ and $z_5$ is unknown. With respect to the coefficient $\gamma$, inequality (19) must be considered similarly to inequality (16) regarding the coefficient $\alpha$. In this case, the right-hand sides of inequalities (16) and (19) are denoted by $\delta_1$ and $\delta_2$, respectively. Let $\delta = \max\{\delta_1, \delta_2\}$, then the condition $(p_{13} p_{34} + p_{14}) ET_4 > \delta$ and formulas (15), (16), (18) and (19) imply the inequalities $\alpha < 0$, $\gamma < 0$. From this, the following conclusion can be made:

**Conclusion 1.** *If $p_{34} > [\delta/(ET_4 - p_{14})/p_{13}]$, then the inequalities $\alpha < 0$, $\gamma < 0$ are true.*

*4.2. The Maximum of the Criterion Function—The Distributions of the Random Variable of the IFR Classes and MTFR*

In this subsection of the paper, sufficient conditions for the existence of the maximum of the Criterion Function (10) will be formulated in two cases. In the first case, the considerations apply to a class of random variable distributions for which the time to failure of a technical object $T_1$ is assumed to be a random variable with increasing damage intensity function $\lambda 1(t)$, i.e., $T_1 \in$ IFR (Increasing Failure Rate). In the second case, a class of random variable distributions with a unimodal failure intensity function, the $T_1 \in$ MTFR (Mean Time to Failure or Repair), is considered. The results of testing the properties of the random variable distributions of the MTFR class are presented in detail in the papers [32–34].

**Conclusion 2.** *If $T_1 \in$ IFR, $\lambda_1(t)$ is differentiable, $\alpha < 0$, $\beta > 0$, $\gamma < 0$, $\beta + \gamma f_1(0+) > 0$, $\lambda_1(\infty) \alpha ET_1 + \beta - \alpha < 0$, then the criterion function $g(x)$ reaches its maximum value.*

**Proof of Conclusion 2.** The derivative of the criterion function $g(x)$ has the form

$$g'(x) = \{\alpha[f_1(x)ET_1(x) - R_1(x)F_1(x)] + \beta R_1(x) + \gamma f_1(x)\}/M^2(x)$$

where M(x) is the denominator of the criterion function g(x).

It is known, that if the time to failure $T_1$ belongs to the class of distributions of the random variable MTFR, then the equality $H(x) = \lambda_1(x) ET_1(x) - F_1(x) \geq 0$ for $x \geq 0$ is true. The class of distributions of the random variable MTFR has been studied in the papers [33,34]. Some lifetime distributions with unimodal damage intensity function belong to the class of MTFR [33,34]. From the fact, that the derivative $H'(x) = \lambda_{1'}(x) ET_1(x)$, it follows that if the damage intensity function $\lambda_1(t)$ increases, the function H(x) also increases. The class of distributions of a random variable with a non-decreasing damage intensity function (IFR) is contained in the MTFR class. The sign of the derivative is the same as the sign of the function

$$h(x) = \alpha[\lambda_1(x)ET_1(x) - F_1(x)] + \beta + \gamma\lambda_1(x)$$

It is known, that $H(0+) = 0$, hence $h(0+) = \beta + \gamma\, f_1(0+) > 0$. From the fact that $\alpha < 0$, $\beta > 0$, $\gamma < 0$ and the function H(x) increases, it follows that the function h(x) decreases from the value $h(0+) = \beta + \gamma\, f_1(0+) > 0$ to the value $h(\infty) = \lambda 1(\infty)\, \alpha\, ET_1 + \beta - \alpha < 0$. It follows from this that the derivative of g'(x) changes sign exactly once from "+" to "−". Hence, it is concluded that the criterion function g(x) reaches exactly one maximum.

If $\lambda_1(\infty) = \infty$, then the following conditions suffice for the existence of the maximum of the criterion function g(x): $T_1 \in$ IFR, differentiability of $\lambda_1(t)$, $\alpha < 0$, $\beta > 0$, $\gamma < 0$, $\beta + \gamma\, f_1(0+) > 0$. An example of such a distribution of a random variable is a Weibull distribution with an increasing damage intensity function. □

From the conclusions 1 and 2, the following sufficient condition for the existence of a maximum of the criterion function follows:

**Conclusion 3.** *If $T_1 \in$ IFR, $\lambda_1(t)$ is differentiable, $\beta + \gamma\, \lambda_1(0+) > 0$, $p_{34} > [\delta/(ET_4 - p_{14})/p_{13}]$, $\lambda_1(\infty)\, \alpha\, ET_1 + \beta - \alpha < 0$, then the criterion function g(x) reaches the maximum value.*

A sufficient condition for the existence of the asymptotic maximum of the availability factor is formulated below. To obtain the availability factor from the criterion function g(x), it suffices to assume the following conditions: $z_1 = 1$, $z_2 = z_3 = z_4 = z_5 = 0$. After considering these conditions in formula (10), $B_1 = 0$, $C_1 = 0$ are obtained. Hence, based on (6), (7) and (9) for $\alpha$, $\beta$, $\gamma$, one can calculate:

$$\alpha = -B = -p_{12}ET_2 - p_{13}ET_3 - (p_{13} + p_{34} + p_{14})ET_4 + (p_{12} + p_{13} + p_{14})ET_5$$
$$\beta = C = ET_5\, ;\ \beta > 0$$
$$\gamma = 0$$

The inequality $\alpha < 0$ is equivalent to the inequality:

$$p_{34} > \left[p_{12}(ET_5 - ET_2) + p_{13}(ET_5 - ET_3) + p_{14}(ET_5 - ET_4)\right]/p_{13}ET_4$$

Given that $\beta > 0$ and $\gamma = 0$, one can now formulate a sufficient condition for the existence of a maximum of the availability factor.

**Conclusion 4.** *If $T_1 \in$ IFR, $\lambda_1(t)$ is differentiable, $\lambda_1(\infty)\, \alpha\, ET_1 + \beta - \alpha < 0$, $p_{34} > [p_{12}\, (ET_5 - ET_2) + p_{13}\, (ET_5 - ET_3) + p_{14}\, (ET_5 - ET_4)]/p_{13}\, ET_4$, then the availability factor reaches exactly one maximum value.*

**Proof of Conclusion 4.** For the availability factor, the derivative of the criterion function has the form:

$$g'(x) = \{\alpha[f_1(x)ET_1(x) - R_1(x)F_1(x)] + \beta R_1(x)\}/M^2(x)$$

where M(x) is the denominator of the criterion function g(x).

If the damage intensity function $\lambda_1(t)$ is increasing, then the function H(x) is increasing. The sign of the derivative is the same as the sign of the function:

$$h(x) = \alpha[\lambda_1(x)ET_1(x) - F_1(x)] + \beta$$

It is known that H(0+) = 0, hence h(0+) = $\beta$ > 0.

From the fact that $p_{34} > [p_{12}$ (ET$_5$–ET$_2$) + $p_{13}$ (ET$_5$–ET$_3$) + $p_{14}$ (ET$_5$–ET$_4$)]/$p_{13}$ ET$_4$, it follows that $\alpha$ < 0 and the function h(x) decreases from h(0+) = $\beta$ > 0 to h($\infty$). If h($\infty$) = $\lambda1(\infty)$ $\alpha$ ET1 + $\beta$ − $\alpha$ < 0, it means that the derivative g′(x) changes sign exactly once from "+" to "−". Hence, it is concluded, that the availability factor g(x) reaches exactly one maximum.

If $\lambda1(\infty) = \infty$, then the following conditions are sufficient for the existence of a maximum of the availability factor:

$$T_1 \in iRF, \ p_{34} > \left[p_{12}(ET_5 - ET_2) + p_{13}(ET_5 - ET_3) + p_{14}(ET_5 - ET_4)\right]/p_{13}ET_4$$

□

## 5. Exemplary Calculation Results

**Example 1.** *In Figure 2, the plots of the criterion function g(x) are shown when g(x) represents profit per unit time, and in Figure 3, when g(x) represents availability for transportation tasks.*

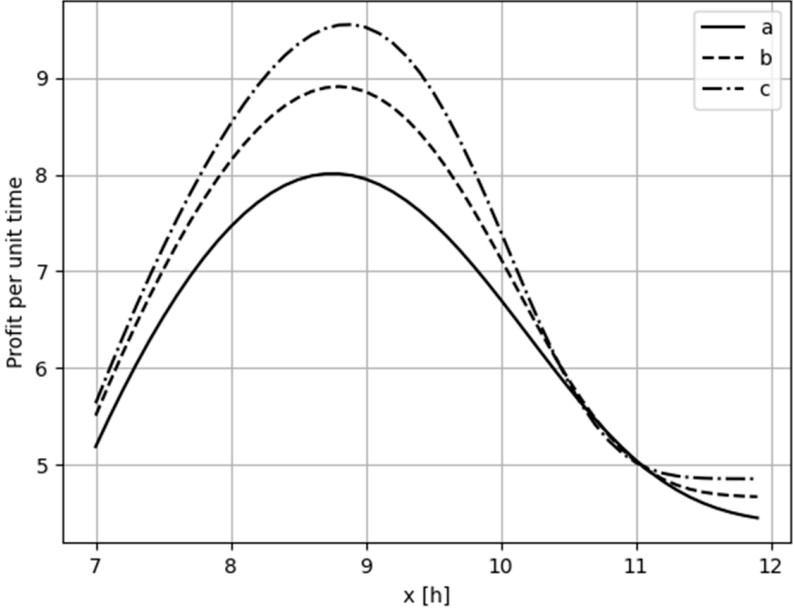

**Figure 2.** The graphs of the function g(x)—profit per unit time as a function of time to preventive maintenance x [h], determined for the Weibull distribution with the following parameter values: scale = 10 and shape = 9 (curve a), shape = 11.5 (curve b), shape = 14 (curve c), the values of the parameters of the distribution of the random variable ET1 were determined for the considered values of the service life of the tested city buses, respectively, 9.25, 9.5, 9.75 [h].

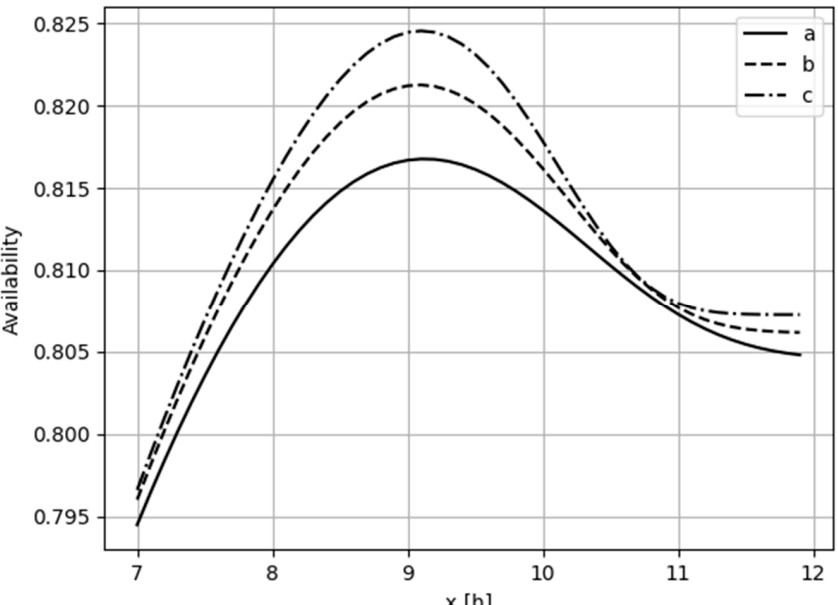

**Figure 3.** The graphs of the function g(x)—availability to perform transport tasks as a function of time to preventive maintenance x [h], determined for the Weibull distribution with the following parameter values: scale = 10 and shape = 9 (curve a), shape = 11.5 (curve b), shape = 14 (curve c), the values of the parameters of the distribution of the random variable ET1 were determined for the considered values of the service life of the tested city buses, respectively, 9.25, 9.5, 9.75 [h].

The calculations were conducted for the following data:

(1) values of the matrix of probabilities of changes of states of the model P:

$$P = \begin{bmatrix} 0 & 0.2033 & 0.0811 & 0.4124 & 0.3032 \\ 1 & 0 & 0 & 0 & 0 \\ 0.7858 & 0 & 0 & 0.2142 & 0 \\ 1 & 0 & 0 & 0 & 0 \\ 1 & 0 & 0 & 0 & 0 \end{bmatrix}$$

(2) the average values of the technical object residence times in the model states in [h]: $ET_2 = 0.389$, $ET_3 = 1.538$, $ET_4 = 3.621$, $ET_5 = 1.783$; for the airworthiness time (time to failure) $ET_1$, a Weibull distribution was assumed for which the value of the scale parameter = 10; three cases were analysed when the value of the shape parameter of the Weibull distribution is, respectively shape $\in \{9, 11.5, 14\}$;

(3) the average values of profits (costs) per unit time in each state of the model in [PLN/h]: $z_1 = 38$, $z_2 = -71$, $z_3 = -117$, $z_4 = -143$, $z_5 = -121$. Both when the criterion function g(x) denotes the profit per unit time (Figure 2) and when g(x) denotes the availability to carry out transport tasks (Figure 3) the criterion function reaches its maximum value. In each of the three cases analysed, for particular values of the shape parameter of the Weibull distribution, there is an optimal value of time to preventive replacement x [h]. Based on the analysis of the value of $x_{max}$, for which the criterion function g(x) reaches its maximum value, it can be concluded that as the value of the shape parameter increases, the value of $x_{max}$ and the maximum value of the criterion function g(x).

**Example 2.** *Figures 4 and 5 show, respectively, the graphs of the criterion function g(x) in the case where g(x) represents profit per time unit and where g(x) represents availability to complete transportation tasks. The calculations were performed for the data of Example 1, assuming that the uptime (time to failure) $ET_1$ has a Weibull distribution, for which the value of the scale*

parameter = 10 and the shape parameter = 11.5. The graphs show four cases: case d—when the number of preventive replacements is the same as in Example 1, and cases a, b, and c, when the number of preventive replacements is increased by 10%, 20%, and 30%, respectively, with respect to case d.

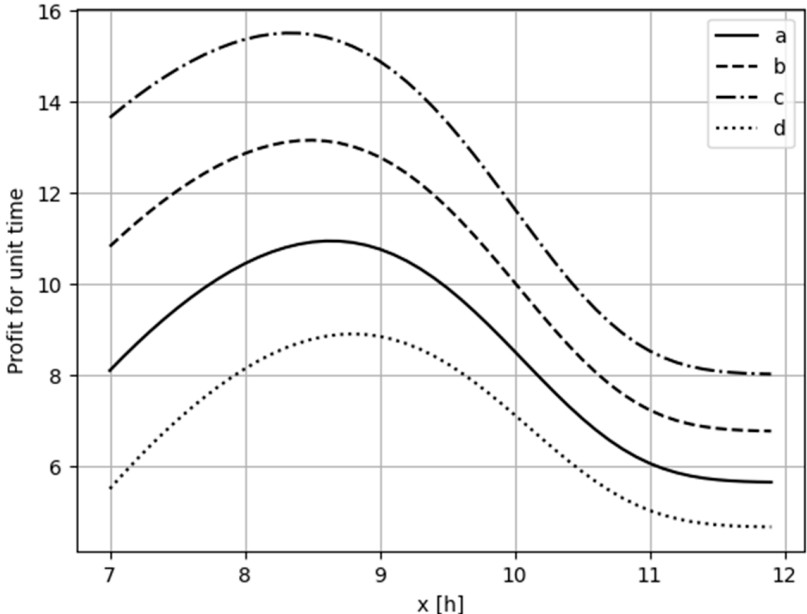

**Figure 4.** The graphs of the function g(x)—profit per time unit as a function of time to preventive replacement x [h], determined when the number of preventive replacements is as in Example 1 (curve d) and when the number of preventive replacements is increased by 10%, 20%, and 30%, respectively (curves a, b, and c).

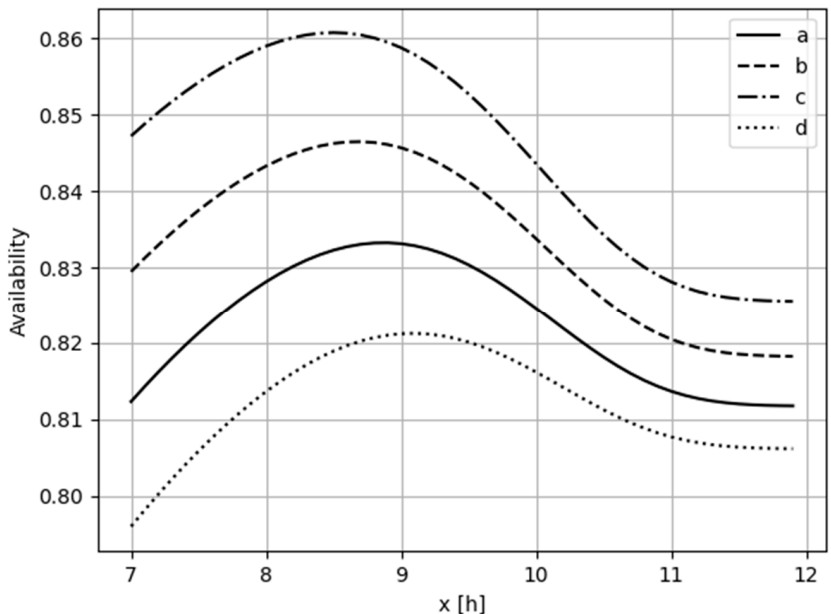

**Figure 5.** Graphs of the function g(x)—availability to perform transportation tasks as a function of time to preventive replacement x [h], determined when the number of preventive replacements is as in example 1 (curve d) and when the number of preventive replacements is increased by 10%, 20%, and 30%, respectively (curves a, b, and c).

Based on the analysis of the graphs presented in Figures 4 and 5, it can be seen that as the number of performed preventive exchanges increases (their frequency in-

creases), the value of the criterion function g(x) increases, both in the case of profit per time unit and availability to perform transport tasks, and the maximum value of the function g(x) is reached for the increasingly smaller values of $x_{max}$ (the optimum time value for preventive replacement).

## 6. Conclusions

The mathematical model presented in the article makes it possible to determine the optimum values of the preventive replacement time in such a way that the criterion functions (profit per unit time and readiness to carry out transport tasks) reached the maximum values. On the basis of the analysis of the results obtained, it can be noted that for the considered input data of the model, the increase in the value of the serviceability time of the examined city buses (increase in the value of the shape parameter of the Weibull distribution) causes an increase in the value of the considered criterion functions (both profit per time unit and readiness), while increasing the optimum time to preventive replacement (Figures 2 and 3, respectively). Decreasing the time to preventive replacement (increasing the frequency of preventive replacements) by 10, 20 and 30%, causes a significant increase in the values of the criterion functions: profit per unit time (from 8.9 to 15.5 [PLN/h]) and readiness to carry out transport tasks from 0.821 to above 0.86 (Figures 4 and 5, respectively). In the paper, the criterion functions are considered over an infinite time horizon. The formulation of stronger conditions requires the establishment of relations between the average stay times of a technical object and the unit profits (costs) in the states of repair by the Technical Emergency Service with the loss of course (state 3) and the preventive replacement (state 5). It has been proved that under general assumptions the criterion functions considered in the paper have exactly one extremum (maximum). On the basis of the conducted analysis, sufficient conditions were formulated for the existence of the maximum of these functions when the time to failure of a technical object is a random variable with an increasing damage intensity function. The assumptions adopted in the model and the formulated conditions define the relations between the input parameters of the developed model and verify the possibility of applying a specific set of input data for determining the optimum preventive replacement times (determining the maximum of the criterion functions). The presented research results constitute the next stage of works on modelling the exploitation systems of technical objects, in which preventive replacements by age are carried out. In the next stages, the models of preventive replacements will be developed for technical objects of other classes than the means of transport, e.g., for power equipment. These models will use both the criterion functions describing economic efficiency (e.g., profit per unit time), operational and technical efficiency (e.g., readiness) and safety (e.g., risk of loss). On this basis, it is planned to develop a comprehensive control method for the subsystem for ensuring the serviceability of technical objects using decision-making semi-Markov processes and non-deterministic methods for determining optimal (sub-optimal) solutions.

**Author Contributions:** Conceptualization, K.M. and S.B.; methodology, K.M. and A.S.; software, A.N.; validation, K.M., investigation, S.B. and A.S.; data curation, S.B. and A.S.; writing—original draft preparation, K.M., writing—review and editing, S.B.; visualization, A.S. and A.N.; supervision, K.M. All authors have read and agreed to the published version of the manuscript.

**Funding:** This research received no external funding.

**Institutional Review Board Statement:** Not applicable.

**Informed Consent Statement:** Not applicable.

**Data Availability Statement:** Not applicable.

**Conflicts of Interest:** The authors declare no conflict of interest.

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
