# Peer review of "Semi-Markov Model of the System of Repairs and Preventive Replacements by Age of City Buses"

_applsci, doi:10.3390/app112110411_

Round 1

Reviewer 1 Report

  • The literature review is too narrow and does not cover all areas of the topic. The literature sources are too old (some of them from 1978y.)

  • Line 58-59 and 68-69:

“The concept of minimal repair was introduced by Brown and Prochan in their work  [3]“ – 1983y after used sentence „However, only one paper quoted there [4] uses semi-Markov processes“ and used literature source was older – 1978y.

  • It is used the wrong format of the literature review. In the scientific literature, it is not enough just to write that some works have been done and show the source. It is necessary to provide analysis on foundings:
    • Line 66-68:

„A review of the used methods of construction of criterion functions in models of minimal repairs with preventive maintenance according to age is provided in works [1, 2]“ – please provide at least the main conclusions from this review.

  • Line 69-70:

“A more recent review of works on minimal repairs is given in the book [5]“  – please provide at least a minimum review of it.

  • Line 78-79:

“This approach to the construction of the criterion function was used in the works [13, 1]“  - how it was used? ...conclusions

  • Line 81 the language should be checked “ are not neligable”.
  • The aim of the research is not clearly presented.
  • The is no description of the used methodology
  • Chapter 3 has just one reference to the source. I doubt that all other formulas were created by the authors of this article and never were described before.
  • How the subchapter 3.1 appears in chapter 4? Line 292
  • Line 415-416: The statement “At the same time it can be stated that the application of semi-Markov processes to modelling of such systems is little“ is not proved by literature review.
  • Line 426-428:

On the basis of the conducted analysis, sufficient conditions for the existence of the maximum of these functions  were formulated.“ It is a fact, not a conclusion. The conclusion would describe the formulated functions

There are no discussions about what this research changing in the current situation and where the findings could be used.

  • Some requirements to the authors are left in the text. Line 435-444

Author Response

1) The literature review is too narrow and does not cover all areas of the topic. The literature sources are too old (some of them from 1978y.)

The literature review has been changed. The article partially contained the content of the article Knopik L, Migawa K. Semi-Markov system model for minimal repair maintenance. Eksploatacja i Niezawodnosc - Maintenance and Reliability 2019; 21 (2): 256-260, http://dx.doi.org/10.17531/ein.2019.2.9. This situation was caused by an error at the stage of translation.

Publications that are judged to be old are, in our opinion, key publications for the issue.

2) Line 58-59 and 68-69:

“The concept of minimal repair was introduced by Brown and Prochan in their work  [3]“ – 1983y after used sentence „However, only one paper quoted there [4] uses semi-Markov processes“ and used literature source was older – 1978y.

The concept of minimal repair can be found in the work of Morse (Morse P.C. 1958), but it was not used in the theory of reliability. As part of improving the theoretical introduction, the disputed fragment was corrected.

  • Line 66-68:

„A review of the used methods of construction of criterion functions in models of minimal repairs with preventive maintenance according to age is provided in works [1, 2]“ – please provide at least the main conclusions from this review.

Completed in accordance with the reviewer's comment

  • Line 69-70:

“A more recent review of works on minimal repairs is given in the book [5]“  – please provide at least a minimum review of it.

Completed in accordance with the reviewer's comment

  • Line 78-79:

“This approach to the construction of the criterion function was used in the works [13, 1]“  - how it was used? ...conclusions

Completed in accordance with the reviewer's comment

  • Line 81 the language should be checked “ are not neligable”.

Completed in accordance with the reviewer's comment

  • The aim of the research is not clearly presented.

Completed in accordance with the reviewer's comment

  • The is no description of the used methodology

The paper contains a mathematical model for determining the optimal replacement strategy by age (the model for determining the optimal preventive replacement time). Based on the identification of the actual system of city bus operation under study, a graph was developed to map the modeled process. The mathematical model of the considered process was developed using the theory of stochastic processes - semi-Markov processes. As a result of solving the systems of linear equations, the stationary distribution of the Markov chain put into the process was determined, and then, using the limit theorem for semi-Markov processes, the limit distribution of this process was determined. On this basis and for the adopted assumptions, the following criteria functions were determined: profit per unit of time and readiness. Then, the conditions for the existence of the maximum of the criterion function were determined in the case when the time to failure of the technical object is a random variable with an increasing failure intensity function. Based on the data obtained from the database of the actual city bus operation system, the input values ​​of the developed model were determined: the probabilities of transitions between the model states, the average duration of the model states and the average unit costs in these states. The input data concerned 182 buses for the period of 12 months of 2016. Both in the case of profit per unit of time and readiness, the test results were developed for the considered values ​​of the serviceability time of technical objects with the Weibull distribution. Additionally, an analysis of the examined criteria functions was carried out, when the number (frequency) of preventive replacements would be increased by 10%, 20% and 30%, respectively.

  • Chapter 3 has just one reference to the source. I doubt that all other formulas were created by the authors of this article and never were described before.

Literature has been added to Chapter 3. The formulas developed for the semi-mark process model are developed by the authors for the manuscript. Formulas (11) and (13) are basic formulas for the theory of reliability and the theory of semi-mark processes

  • How the subchapter 3.1 appears in chapter 4? Line 292

Editing error. Changed in order.

  • Line 415-416: The statement “At the same time it can be stated that the application of semi-Markov processes to modelling of such systems is little“ is not proved by literature review.

Removed

  • Line 426-428:

On the basis of the conducted analysis, sufficient conditions for the existence of the maximum of these functions were formulated.“ It is a fact, not a conclusion. The conclusion would describe the formulated functions

There are no discussions about what this research changing in the current situation and where the findings could be used.

Completed. The formulated conditions define and verify the possibility of using a specific set of input data to determine the extremes of the criterion functions (determining the optimal preventive replacement times)

  • Some requirements to the authors are left in the text. Line 435-444

Deleted

Reviewer 2 Report

1. The abbreviations were recommended to be presented with its full name when they first appeared in the manuscript, such as PM, CM in the introduction section. 2. The authors were recommended to polish the manuscript for the purpose of avoiding potential grammar mistakes. For instance, “….various preventive action strategies are introduced into systems management. ” 3. What’s the difference between semi-markov process and full markov procedure in your study. 4. Please explain the legend a, b, c in the figure 2 with more details. 5. The following studies were recommended to properly cited [1] [1] X. Chen, H. Chen, Y. Yang, H. Wu, W. Zhang, J. Zhao, et al., Traffic flow prediction by an ensemble framework with data denoising and deep learning model, Physica A: Statistical Mechanics and its Applications, vol. 565, p. 125574, 2021. [2] Deep Learning-Based Energy Management of an All-Electric City Bus With Wireless Power Transfer, IEEE Access, vol. 9, pp. 43981-43990, 2021.

Author Response

  1. The abbreviations were recommended to be presented with its full name when they first appeared in the manuscript, such as PM, CM in the introduction section.

Corrected in accordance with the reviewer's comments

  1. The authors were recommended to polish the manuscript for the purpose of avoiding potential grammar mistakes. For instance, “….various preventive action strategies are introduced into systems management. ”

Linguistic correction was made in accordance with the comments of the reviewer

  1. What’s the difference between semi-markov process and full markov procedure in your study.

In the case of the Markov process, it is assumed that the distributions of the considered random variables - residence times in individual states of the modeled process - have an exponential distribution. This is a significant limitation because it is practically impossible to meet this assumption in real systems. In the case of semi-Markov processes, such a condition does not have to be met. This means that the residence times in the states of the modeled process can be described by any type of distribution.

  1. Please explain the legend a, b, c in the figure 2 with more details.

Corrected in accordance with the reviewer's comments

  1. The following studies were recommended to properly cited [1] [1] X. Chen, H. Chen, Y. Yang, H. Wu, W. Zhang, J. Zhao, et al., Traffic flow prediction by an ensemble framework with data denoising and deep learning model, Physica A: Statistical Mechanics and its Applications, vol. 565, p. 125574, 2021. [2] Deep Learning-Based Energy Management of an All-Electric City Bus With Wireless Power Transfer, IEEE Access, vol. 9, pp. 43981-43990, 2021.

Corrected in accordance with the reviewer's comments

Reviewer 3 Report

  • The first chapter is a self-plagiarism of authors from article: Knopik L, Migawa K. Semi-Markov system model for minimal repair maintenance. Eksploatacja i Niezawodnosc – Maintenance and Reliability 2019; 21 (2): 256–260, http://dx.doi.org/10.17531/ein.2019.2.9.
  • Even figure is the same as long article is too similar, I recommend rejecting to publication as long this article will too similar to previous one.

Author Response

    The first chapter is a self-plagiarism of authors from article: Knopik L, Migawa K. Semi-Markov system model for minimal repair maintenance. Eksploatacja i Niezawodnosc – Maintenance and Reliability 2019; 21 (2): 256–260, http://dx.doi.org/10.17531/ein.2019.2.9.

    Even figure is the same as long article is too similar, I recommend rejecting to publication as long this article will too similar to previous one.

The literature review has been changed. The article partially contained the content of the article Knopik L, Migawa K. Semi-Markov system model for minimal repair maintenance. Eksploatacja i Niezawodnosc - Maintenance and Reliability 2019; 21 (2): 256-260, http://dx.doi.org/10.17531/ein.2019.2.9. This situation was caused by an error at the stage of translation.

This manuscript is a case-by-case analysis of bus transport. The article "Semi-Markov system model for minimal repair maintenance" contains basic descriptions of the issue, and therefore some of its components are similar to the currently published one. A detailed description of the changes is included in the introduction, referring to the publications [......]

The results presented in the Semi-Markov system model for minimal repair maintenance are not compiled on the basis of real data, but data constituting "numerical examples".

The development of assumptions, model and results is an original creative achievement, it is based on this article.

The paper contains a mathematical model for determining the optimal replacement strategy by age (the model for determining the optimal preventive replacement time). Based on the identification of the actual system of city bus operation under study, a graph was developed to map the modeled process. The mathematical model of the considered process was developed using the theory of stochastic processes - semi-Markov processes. As a result of solving the systems of linear equations, the stationary distribution of the Markov chain put into the process was determined, and then, using the limit theorem for semi-Markov processes, the limit distribution of this process was determined. On this basis and for the adopted assumptions, the following criteria functions were determined: profit per unit of time and readiness. Then, the conditions for the existence of the maximum of the criterion function were determined in the case when the time to failure of the technical object is a random variable with an increasing failure intensity function. Based on the data obtained from the database of the actual city bus operation system, the input values of the developed model were determined: the probabilities of transitions between the model states, the average duration of the model states and the average unit costs in these states. The input data concerned 182 buses for the period of 12 months of 2016. Both in the case of profit per unit of time and readiness, the test results were developed for the considered values of the serviceability time of technical objects with the Weibull distribution. Additionally, an analysis of the examined criteria functions was carried out, when the number (frequency) of preventive replacements would be increased by 10%, 20% and 30%, respectively.

Round 2

Reviewer 1 Report

1. Authors:" Formulas (11) and (13) are basic formulas for the theory of reliability and the theory of semi-mark processes"

Still, it is not developed by you and needs to have a literature source.

2. The first part of the conclusions still sounds like a summary of the article, but not conclusions.

Author Response

1. Authors:" Formulas (11) and (13) are basic formulas for the theory of reliability and the theory of semi-mark processes"

Still, it is not developed by you and needs to have a literature source.

We added references. It was simply oblivion on our part. 

2. The first part of the conclusions still sounds like a summary of the article, but not conclusions.

We improved the conclusions. The changes are visible in the file. 

Reviewer 2 Report

My comments have been addressed. 

Author Response

Thank you for helping us improve the manuscript. 

Reviewer 3 Report

Thank you for your answers and for changing chapter 1.

Author Response

Thank you for helping us improve the manuscript. 

We improved the introduction by adding information on the subject of transport. The changes are visible in the manuscript.